# Intelligent Materials for Labeling Dentures in Forensic Dental Identification—A Pilot Study

**Corina Laura Ştefănescu** [1,*], **Lavinia Simona Neculai-Cândea** [2], **Marius Florentin Popa** [2], **Agripina Zaharia** [1], **Rodica Maria Murineanu** [1], **Ştefan Pricop** [2], **Liliana Sachelarie** [3,*], **Loredana Liliana Hurjui** [4] **and Vlad Danila** [4]

1 Faculty of Dental Medicine, "Ovidius" University Constanta, 124, Bvld. Mamaia, 900527 Constanta, Romania; agrizaharia@yahoo.com (A.Z.)
2 Faculty of Medicine, "Ovidius" University Constanta, 124, Bvld. Mamaia, 900527 Constanta, Romania; lavinia_candea@ymail.com (L.S.N.-C.); marius_popa2005@yahoo.com (M.F.P.)
3 Faculty of Dental Medicine, "Appolonia" University, 11, Păcurari Street, 700511 Iasi, Romania
4 Department of Medical Disciplines, Faculty of Dental Medicine, "Grigore T. Popa" University of Medicine and Pharmacy, 700115 Iasi, Romania
* Correspondence: corina.stefanescu@yahoo.com (C.L.Ş.); lisachero@yahoo.com (L.S.)

**Abstract:** (1) Background: the present study continues previous research on new marking techniques by using intelligent materials, NFC-tags (Near Field Communication tags) and Aerogel. They prevent the loss of information over time for cadavers of unknown identities in various stages of destruction or unknown living victims, as well as missing persons who have lost or are concealing their identity. (2) Methods: this study consisted of a technological and an experimental stage. In the technological stage, two different sizes were used: round (Ø 1 cm, 0.1 mm thickness, 0.1 g weight) and square (5/5/0.1 mm), both with a 140 byte memory and high-temperature resistance (at max. 200 °C), (by the classical producing technology). After loading the identification information on the NFC-tags, they were embedded (either alone or protected by Aerogel) in the sample dentures using a new "sandwich technique" method, before the polymerization process. In the experimental stage, the sample dentures with the new materials were exposed to various damaging environments such as liquid media (sea water, fresh water, alcohol 40%, and HCl 0.2%) in order to test the time resistance of the identification elements. The samples were monitored and tested over four years. (3) Results: the information stored on the NFC tags was retrieved unaltered at the end of the monitoring period, regardless of the damage caused by the liquid media to the sample denture material and will provide an innovative solution as compared to other labeling methods. (4) Conclusion: the use of intelligent materials for labeling acrylic dentures provides additional reliability by preserving the identification information over time.

**Keywords:** labeling dentures; NFC-tag; Aerogel; forensic odontology; identification

## 1. Introduction

The dental system is a morpho-structural complex consisting of a range of units defined in number, shape, and size during developmental periods, and at the same time it is endowed with several individual normal and pathological morphological features. These are supplemented by the permanent clinical and radiological features of specific treatments, all of which make it possible to establish an individualized dental formula [1].

Forensic odontology is an integral part of forensic science, using dental or orofacial findings to serve the judicial system, and it involves close collaboration between forensic pathologists, anthropologists, and dentists [2]. It assumes a primary role in identifying the remains of corpses when they are affected by post-mortem changes (putrefaction, terrestrial and aquatic fauna, etc.), by the complex lesion picture that alters the face, or simply when the person's fingerprints are missing. Any dentist can provide, in legal cases and by a special request issued by entitled authorities, data from his patients' files which can

lead to their identification, such as X-rays, individual aspects of the dentition, dental and maxillary anomalies, and a history of various treatments and prosthetic works. Therefore, it is essential that dentists keep accurate dental records so that, if needed, all information regarding malpractice, negligence, fraud, or abuse, as well as the identity of unknown persons, can be revealed [2–4].

On the other hand, dentists need special training in the methods and techniques used in forensic odontology to be able to participate, together with specialists in anthropology and forensic pathology, in the identification process of people or bodies [5–7].

Moreover, the unique anatomical aspect of dentition and customized restorations ensure accuracy in forensic identification techniques. Bite mark and maxillofacial trauma analysis play an important role in human identification [8–11].

In dental practice, the notions of general toxicology are important in diagnosis and therapeutic implications. The oral cavity is both an entry gate and a place for the release of toxic substances and traces of acute or chronic intoxications that are of major importance in the forensic field and in that of diagnosis and dental interpretation [12,13]. Special attention is paid to trauma caused by chemical agents [14].

Looking back, there is a history of identifying people using prosthetic works [11]. This history began in 1835 when the Countess of Salisbury was identified by her gold dentures. In 1850, in the case of Dr. Parkman, the identification was made using bone fragments and segments of his mobile dentures. In 1972, 27 people were identified in the Houston mass murders using dental evidence. In 1976, after the Thomson Grand Canyon flood in Colorado, 139 bodies were recovered and identified using computer-aided dental identification for the first time. Three years later, in 1979, following the Chicago and San Diego plane crashes, 191 bodies were identified through dental records [11]. In 2011, after the September 11 terrorist attack, under the coordination of Dr. Jeffrey Burkes, a chief forensic dental consultant from New York City, 40 dental specialists and 100 dental volunteers worked to identify the victims. After the 2015 terrorist attack in Paris, of the 41 unidentified corpses, 22 were identified with the collaboration of dentists [12].

The marking of dentures has been extensively researched. The inclusion of personal identification elements in different types of prosthetic works is achieved via various methods: inscription, engraving, or incorporation. Labeling using these methods was, at a time, being done after the completion of denture manufacturing.. However, they all showed lack of durability over time due to the action of internal or external factors which negatively affected the acrylate resin [13,14].

NFC-tags (Near Field Communication -tags) are devices of different sizes which allow short range wireless communication through their components. They consist of a copper coil and a microchip. Depending on its internal memory, the microchip allows for a larger or smaller amount of data storage. Through electromagnetic induction, the coil allows for reading of the information stored on the microchip using a NFC reader, a common application present on smartphones [15–17].

Aerogel is a synthetic chemical and a nanostructured material, and is included in the category of intelligent materials.

Depending on its compounds and the production method, it comes in two forms: rigid foam (Figure 1a) or granules (silica Aerogels) (Figure 1b).

The studies on the physical and chemical properties of Aerogel have concluded that it is a very good thermal insulator, has a high degree of absorption of infrared radiation, and a very high melting point (1500 °C).

Its uses are seen to be almost infinite in fields such as industry, construction, aeronautics, ecology, etc. Silica aerogels have uses in many fields: fabric insulation (from sports equipment to astronaut suits); construction (from window insulation to increasing the energy efficiency of buildings); corrosion prevention (from small equipment to aeronautical and space industry); and functional packaging (in the food industry) [17]. Therefore, Aerogel has been called the "wonder material of the future" and the "ecofriendly synthetic chemical of the century" [18–22].

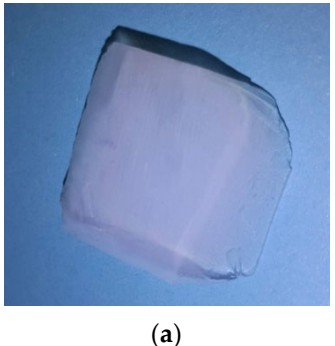
(**a**)

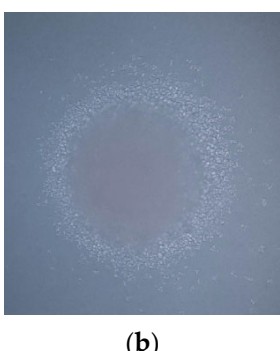
(**b**)

**Figure 1.** (**a**)—Aerogel–rigid foam, (**b**)—Granules (silica Aerogel).

In this work, the properties of these intelligent materials (NFC-tags and Aerogel) were studied and tested for their ability to preserve certain identification data over time, unaltered, taking advantage of Aerogel's insulation ability.

The objectives of this study are: (1) to introduce new methods of incorporating intelligent materials (NFC-tags and Aerogel) in the technological process of making acrylic dentures; and (2) these labeling methods provide reliability and durability, over time, of the embedded information for the purpose of forensic identification.

## 2. Materials and Methods

The main stages of our research were: (a) producing "sample dentures"; (b) embedding the intelligent materials; and (c) exposing the sample dentures to various damaging environments in order to test the time resistance of identification elements.

### 2.1. Materials

2.1.1. Materials for Making Sample Dentures

For our research, a total of 32 rectangular sample dentures were made. The materials used for their manufacturing were identical to those used in making standard acrylic complete dentures: heat-curing acrylic resin PMMA (polymethyl methacrylate) (liquid monomer, polymer powder) in two forms, pink (Meliodent HC 1000 g 3—Pink, Heraeus Kulzer, Germany) and transparent acrylic resin (Meliodent HC 1000 g 1—Clear, Heraeus Kulzer, Germany); modeling wax (Ceradent, Spofa Dental, Czech Republic); flasks for investing and processing the samples; a boiling water bath; an insulating substance (Isodent, Spofa Dental, Czech Republic); packing plasters (Hera Moldano, Kulzer, Mitsui Chemical Group); a hydraulic press; milling cutters (Bredent, Germany) for processing; finishing compounds (Bredent, Germany); and a felt wheel (Bredent, Germany) for polishing to a high gloss. The technological process followed all the traditional steps of making the base of a final denture (thermal acrylic polymerization and elimination of wax).

The two types of acrylic materials (pink and transparent) with different composition and properties are commonly used to make complete and partial dentures, as well as overdentures on implants or natural abutments. Both pink and transparent acrylic materials were used to better observe possible differences in the last stage of the investigation resulting from resin contact with the chosen environments.

2.1.2. Intelligent Materials (NFC-Tags, Aerogel)

In this study, NFC-tags of two different sizes were used: (a) round, Ø 1cm, 0.1 mm thickness, 0.1 g weight (Figure 2a); and (Figure 2b) square, 5/5/0.1 mm (Figure 2b), both with 140 bytes of memory and high-temperature resistance (at max. 200 °C). The round NFC-tags were embedded in the pink sample dentures, while the square NFC-tags, due to their smaller size, were embedded in the transparent ones, for aesthetic reasons.

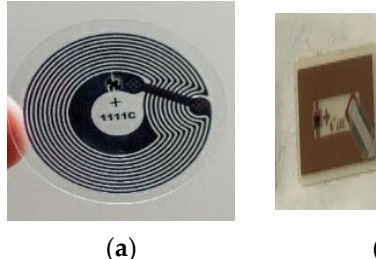 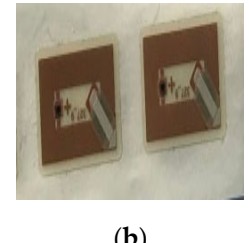

(**a**)　　　　　　　　(**b**)

**Figure 2.** (**a**)—NFC-tag—rounded: Ø 2cm, 0.1 mm thickness, 0.1 g weight; (**b**)—NFC-tag—square: 5/5/0.1 mm.

Given its thermo-resistant properties, Aerogel is intended to provide an extra protective layer for the information that will be stored on thSe NFC-tags. Silica Aerogel in granular form (LUMIRA® LA 1000, Cabot Corporation, Billerica, MA, USA), with a grain size of 100 µm, 1.2 mm was used (Figure 3).

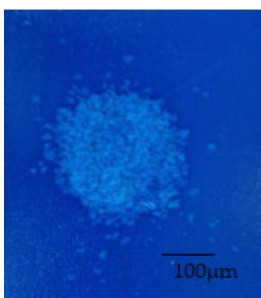

**Figure 3.** Particles of silica Aerogel.

### 2.1.3. Experimental Liquid Media

Given the possible interactions of external or internal conditions on denture wearers, bodies, or people who need possible identification (drowning in sea water or freshwater, people with gastroesophageal reflux disease, and alcoholics), the following liquid materials were chosen: fresh water, sea water, HCl 0.2%, and alcohol 40%. Through prolonged action, these liquids can induce alterations of the denture materials.

### 2.2. Methods

A total of 16 sample dentures made of plain pink acrylic resin (Meliodent HC 1000 g 3—Pink, Heraeus Kulzer, Germany) and 16 sample dentures made of plain transparent acrylic resin (Meliodent HC 1000 g 1—Clear, Heraeus Kulzer, Germany) were made.

### 2.2.1. Making the Mold for Sample Dentures

To make the mold, 2 mm-thick pink denture wax plates were used, which were heated over a Bunsen burner until they were plasticized and became elastic enough to allow for shaping. Rectangular shapes of 2/4/0.2 cm—wax models—were cut out. Denture flasks were used to invest the sample dentures. The flasks had two parts and a cover that fit perfectly. In the flask base, a creamy mixture of gypsum was applied, and two wax models were positioned within. After the plaster set, the following steps were hydration (in cold water for 10 min), and drying and insulation with an alginic solution (Isodent (Spofa Dental, Czech Republic). Afterwards, the upper part of the flask was applied, another amount of gypsum was poured until it was completely filled, and the cover was put into the proper position.

The flask was dipped in a boiling water bath for 8 min to dewax. The two parts of the flasks were separated, thus obtaining the mold. All wax traces were removed, and the mold was degreased with alcohol. After cooling, the insulating alginic solution was applied all over the surface to avoid adhesion of the acrylate resin to the mold.

### 2.2.2. Making Control Samples

To make the control samples (with no intelligent materials), an acrylic paste was prepared by manually mixing polymer powder and the liquid monomer.

The mixture went through several consistencies: sandy, fibrous, doughy, and rubbery. In the dough phase, the acrylic resin was inserted in excess into the mold via manual pressing.

After applying the acrylic paste to the mold, the two parts of the flask were pressed with the hydraulic press, thus removing the excess acrylate. The mold was inserted into a ring to avoid the separation of the two parts during polymerization.

The polymerization process was performed in a water tank, by slowly heating the water up to 100 °C, for a total time of 150 min. The flask was left to cool in water until it reached room temperature. A gypsum knife was used to separate the two parts of the flask. The raw, unfinished sample dentures were finally removed from the mold. Their rough processing was done with milling acrylic burs of different types and sizes to remove the excess acrylic material and plaster. The smoothing and finishing was performed with special rubbers–polypants of different grain widths (Bredent, Germany).

### 2.2.3. Making Samples Embedded with Intelligent Materials

Up to this point, the steps were carried out according to the usual theoretical and practical technical principles of denture manufacturing.

Thus, the finite form of the sample denture was obtained, meeting all the requirements of a complete acrylic denture base, to which the intelligent materials (NFC-tags and Aerogel) were added.

The novel elements of the study began at the stage where the acrylic material was placed in the mold. In this step the intelligent materials were introduced, using the "sandwich" technique (described below), followed by the polymerization process.

Before inserting the NFC-tags into the sample dentures, the first ones were checked for manufacturing defects. Information was entered in the NFC-tags using a smartphone with the NFC Reader application installed.

The information entered in each NFC-tag included three items:

(1)  Type of acrylic material (pink or transparent);
(2)  Type of intelligent material (NFC–tag alone, Aerogel alone, NFC-tag + Aerogel);
(3)  Type of environment to which it will be exposed (sea water, fresh water, alcohol, and chlorohydric acid).

In the acrylate application step, the sandwich technique was used:

-  Three-layer sandwich technique for sample dentures with NFC -tag alone (acrylic layer + NFC-tag + acrylic layer): in the mold, a uniform layer of acrylate was applied, followed by positioning the NFC-tag centrally, and covering everything with a second layer of acrylate.
-  Three-layer sandwich technique for sample dentures with Aerogel alone (acrylic layer + Aerogel + acrylic layer): in the mold, the first layer of acrylate was applied, followed by sprinkling Aerogel granules, and covering everything with the second layer of acrylate.
-  Five-layer sandwich technique for sample dentures with NFC-tag + Aerogel (acrylic layer + Aerogel + NFC-tag + Aerogel + acrylic layer): after applying the first layer of acrylate, a small concavity was created in the center of the mold using digital pressure, in which Aerogel granules were sprinkled with a dental spatula. The NFC-tag was applied, over which a new layer of Aerogel granules was sprinkled. The "sandwich" was finished with the second layer of acrylic paste.

The sample dentures in the experiment were distributed, as show in Table 1.

**Table 1.** Number of sample distribution by acrylic resin type and embedded items.

| Sample Dentures | Control Samples [1] | NFC-Tag Alone [2] | Aerogel Alone [2] | NFC-Tag + Aerogel [3] |
|---|---|---|---|---|
| Pink acrylic resin | 4 | 4 | 4 | 4 |
| Transparent acrylic resin | 4 | 4 | 4 | 4 |

Notes: [1] with no intelligent materials embedded; [2] three-layer sandwich technique; [3] five-layer sandwich techniques.

After polymerization and de-flasking, the sample dentures were processed and polished similarly to common acrylic dentures (Figure 4a–c).

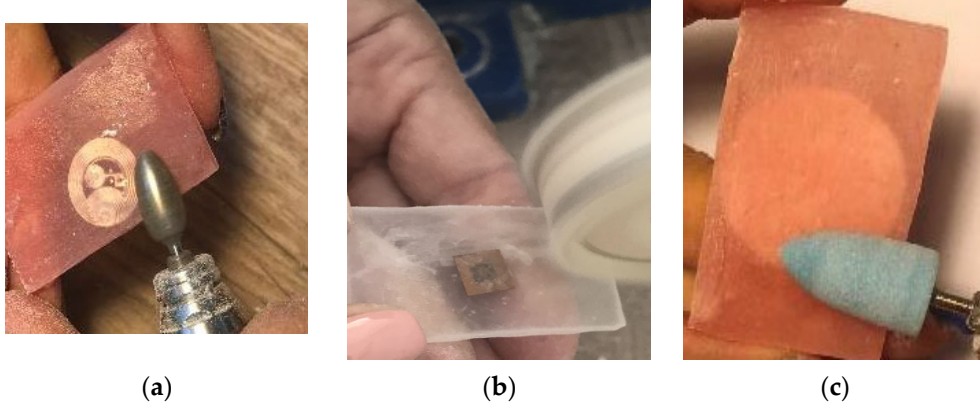

(**a**)                    (**b**)                    (**c**)

**Figure 4.** (**a**) Step in the processing of the pink/NFC-tag alone sample denture; (**b**) Step in the polishing of the transparent/NFC-tag alone sample denture; (**c**) Stage in the finishing of the pink/NFC-tag-Aerogel sample denture.

Thus, the finite form of the sample denture was obtained, meeting all the technical requirements of a complete acrylic denture base in which intelligent materials were embedded.

2.2.4. Experimental Phase

To test their resistance in time, the sample dentures were subsequently experimentally exposed to different environments.

Before immersing the sample denture in the liquid media, all NFC-tags were read in order to confirm the stored information (Figure 5).

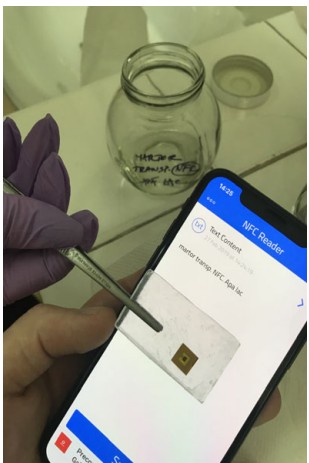

**Figure 5.** Checking stored information before immersion.

In each container, one tested sample was immersed. Samples were stored in closed glass containers, with the liquid completely covering them, at room temperature and day light, for 4 years.

The initial monitoring stage took 3 months. The sample dentures were examined, and NFC-tags read daily during the first week, then weekly during the first month, every 2 weeks during the second month, and finally at the end of the third month.

The examination and readings were performed annually for 4 years. The samples were examined for macroscopic changes in basic properties and read with the smartphone to check whether the intelligent materials retained the stored information under the action of the liquid media.

## 3. Results

The scheme of the experiment is shown in Figure 6.

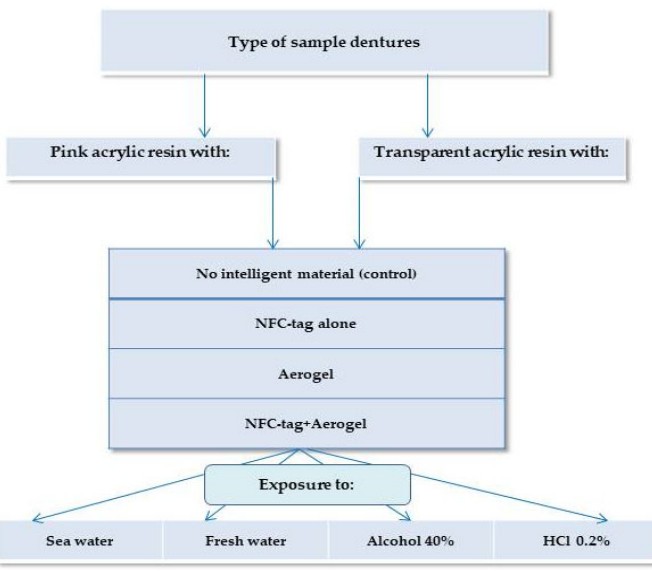

**Figure 6.** Flow work diagram.

The technological stage was the first step towards experimenting with the possible interactions of new materials (NFC-tags; Aerogel) not yet used in dental practice, with conventional acrylic denture materials.

The sample dentures were macroscopically examined after the completion of the technological process; no changes were found in the structure of the acrylic material. The intelligent materials used did not interact with the physical characteristics of the acrylic resin used for denture manufacturing.

A positioning error was observed in three of the sample dentures; the round NFC-tags slid from the initial position towards the edge of the sample by 1–3 mm, although not outside the denture, and could still be read (Figure 7a,b).

The information introduced in the NFC-tags before embedding them in the dentures was retained after the completion of the technological steps. The check-up of the stored information was done by re-reading it after the completion of the manufacturing process. In the monitoring stage, no changes were observed during the first two months of immersion.

In the third month of monitoring, four sample dentures (two containing NFC-tags alone and two containing NFC-tag + Aerogel) placed in a 0.2% HCl solution were no longer readable. All remaining sample dentures were readable and showed no macroscopic structural changes (Table 2).

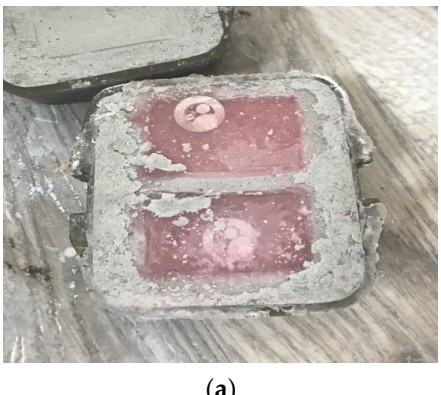 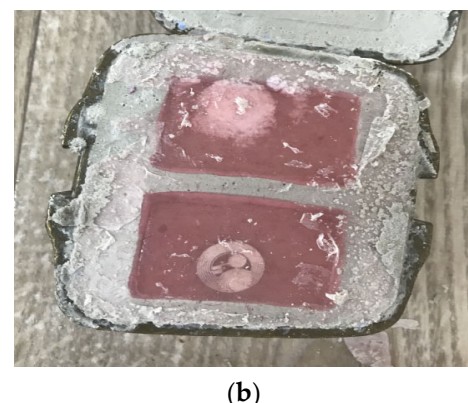

(**a**)                               (**b**)

**Figure 7.** (**a**)—Pink acrylic sample denture marked with NFC-tag alone that migrated from the insertion site; (**b**)—Pink acrylic marked with a NFC-tag and Aerogel that migrated from the insertion site.

**Table 2.** Number of samples read (with the NFC reader application) after 3 months in storage in different environments.

| Sample Type | Liquid Media | | | |
|---|---|---|---|---|
| | Sea Water | Fresh Water | Alcohol (40%) | HCl (0.2%) |
| Acrylic layer/NFC-tag/acrylic layer/PINK | 4/4 | 4/4 | 4/4 | 2 */4 |
| Acrylic layer/NFC-tag/acrylic layer/TRANSPARENT | 4/4 | 4/4 | 4/4 | 4/4 |
| Acrylic layer/Aerogel/NFC-tag/Aerogel/acrylic layer/PINK | 4/4 | 4/4 | 4/4 | 2 */4 |
| Acrylic layer/Aerogel/NFC-tag/Aerogel/acrylic layer/TRANSPARENT | 4/4 | 4/4 | 4/4 | 4/4 |

Note: * Samples are not readable due to an error in the technological stage of manufacturing.

The experiment was extended by replacing the four sample dentures that became unreadable in the third month of exposure to the chlorohydric acid solution with new ones of the same types, and immersing them in 0.2% HCl solution also for 3 months. These new samples were submitted to the same monitoring protocol as the others. All four new samples were still readable after 3 months. Therefore, it was assumed that, in the case of the four sample dentures that failed, most likely there was an error in the technological stage of manufacturing.

The comparative macroscopic analysis of the 32 sample dentures showed that, over longer periods of time, the environments in which they were immersed acted on the acrylic materials by changing certain properties. Starting at the end of the first year of monitoring, the acrylic resin showed imbibition processes, becoming porous and pigmented, in some places opaque (the transparent ones) or rough (the pink ones).

At each monitoring stage, all the sample dentures labeled with a NFC-tag alone and with NFC-tag + Aerogel, embedded in both types of acrylic resins (pink and transparent), showed unaltered readable information.

## 4. Discussion

Previous studies [16–18] concluded that labeling dentures using various methods—surface related or by engraving—have certain limitations, in that they are applied after the completion of the technological stage of denture manufacturing. The information can be lost over time due to the action of external and internal environmental factors to which the body is subjected, and the long-term durability of the information is uncertain [17,18].

Moreover, these labeling methods can create discomfort for patients from a tactile as well as an aesthetic point-of-view. Dentures labeled with NFC-tags alone or embedded in Aerogel provide patient comfort without causing aesthetic or tactile disturbances due to the extremely small size of the NFC-tags and the transparency of the Aerogel, being embedded in visually inaccessible areas [23–25].

Intelligent materials such as NFC-tags and Aerogel have not been studied before from a technological or experimental point-of-view for their use in the medical or forensic field. They were found to be compatible with each other without altering the properties for which they were chosen for in this study.

The experiment showed that NFC-tags are likely to be resistant over time in various environments. Therefore, inputting relevant personal data (name/age/social security number/medical history/address/emergency contact phone number) on the NFC-tag could significantly facilitate efforts in identifying decayed or vastly damaged cadavers [19,20].

There are no data in the literature on the methods or techniques for handling Aerogel or Aerogel in combination with NFC-tags, much less on their incorporation in dental work, which is why this experimental study provides objective new information supported by positive results.

This study on the embedding of these materials in the structure of acrylic dentures, and their exposure for a significant period to various internal and external environments, provides an innovative solution.

Testing the incorporation of these materials (NFC-tags and Aerogel) into dentures at an experimental level for use in forensic odontology identification has been successful and has opened a field for research and implementation of techniques on ways to manipulate these materials into dentures and possibly, in the future, in the very teeth themselves [21,22].

The effectiveness of labeling with NFC-tags alone or NFC-tags embedded in Aerogel layers is demonstrated as early as the technological stage of denture polymerization, in which the physical properties remain unchanged.

Exposure of Aerogel and NFC-tags to environments unsuitable for data preservation even for long periods of time has shown that these materials do not changed their physical properties, thus highlighting the value of their use in the marking of dentures for forensic odontology identification purposes.

There are several possible limitations in implementing the technology we are proposing. The first one is related to the manual inclusion of the NFC-tag in the technological process of making the sample dentures [24]. It can result, during the acrylate pressing phase, in the migration of the NFC-tag from the place where it was initially positioned.

Another one is related to the application phase of the Aerogel [25]. When manually sprinkling the Aerogel (with a grain size between 100 µm–1.2 mm), it can spread irregularly on the surface of the acrylate, resulting in an uneven layer, which can migrate during pressing.

These intelligent materials were used in the present study for the first time in the medical and dental fields. Smart materials are expensive, and in current practice the costs of implementing this technology for labeling dentures are high.

## 5. Conclusions

The effectiveness of marking sample dentures with a NFC-tag alone or with NFC-tag + Aero-gel was demonstrated in this study: the information remained unchanged for a period of four years, regardless of the environment in which they were immersed (that usually damage or destroy other types of labeling) and despite the changes that occurred in the structure of the acrylic resin.

The properties of the smart materials used in this present study demonstrate the possibility of using the "sandwich technique" before the polymerization process. This way, the information is protected inside the sample dentures. Labeling via other methods (inscription, engraving, or incorporation) is done after the completion of the technological

process of denture making, and on their surface; the information being more vulnerable to different environmental conditions.

The small size of NFC-tags and their placement inside dentures in less visible positions (e.g., at the base of palatal plate or in lingual areas of lower denture base) provide comfort for the patient and do not affect the aesthetics.

**Author Contributions:** Conceptualization, C.L.Ş., R.M.M., L.S.N.-C. and M.F.P.; methodology, A.Z.; software, R.M.M.; validation, C.L.Ş., L.S. and V.D.; investigation, C.L.Ş., R.M.M., L.S.N.-C., A.Z. and M.F.P.; resources, M.F.P.; data curation, C.L.Ş.; writing—original draft preparation, Ş.P. and L.S.; writing—review and editing, L.S., L.L.H.; project administration, C.L.Ş. All authors have read and agreed to the published version of the manuscript. All authors contributed equally to this paper as did the first author or the corresponding author.

**Funding:** This research received no external funding.

**Institutional Review Board Statement:** Not applicable.

**Informed Consent Statement:** Not applicable.

**Data Availability Statement:** Not applicable.

**Conflicts of Interest:** The authors declare no conflict of interest.

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
