# Peer review of "Intelligent Materials for Labeling Dentures in Forensic Dental Identification—A Pilot Study"

_applsci, doi:10.3390/app13095574_

Round 1

Reviewer 1 Report (New Reviewer)

Result: Should be more detailed. The NFC tag and aerogel were embedded in the sample denture, and the displacement was generated after processing, and the displacement distance needed to be recorded, and the two different NFC tags were compared at the same time, and the conclusions were organized into tables to make the conclusions clearer.

Author Response

The authors acknowledge the useful observations and suggestions of the reviewer’s as concerns the manuscript entitled: 

Intelligent materials for labeling dentures in forensic dental identification

Corina Laura Åžtefănescu1,*, Lavinia Simona Neculai Cândea2, Marius Florentin Popa2, Agripina Zaharia1, Rodica Maria Murineanu1, Åžtefan Pricop2, Liliana Sachelarie3,*, Loredana Liliana Hurjui4, Vlad Danila4

According to the reviewer’s recommendations, all the suggestions were taken into account, as follows:

    Result: Should be more detailed. The NFC tag and aerogel were embedded in the sample denture, and the displacement was generated after processing, and the displacement distance needed to be recorded, and the two different NFC tags were compared at the same time, and the conclusions were organized into tables to make the conclusions clearer.

All your observations can be found in the new version. Thank you!

After the evaluation of all reviewers, the article was completely redone.

Thank you very much for your review,

Respectfully,

Prof. dr. Liliana Sachelarie

Reviewer 2 Report (New Reviewer)

This paper aimed at providing a novel labeling method on dentures for individual identifications. This is an interesting and meaningful field which is not fully studied yet.   

However, the poor writing, paragraphing and figure layout made the manuscript hard to read. Major comments:

1. I strongly recommend the authors reorganize the paragraphs and the figures, especially the method part. 

2. Another big flaw of this paper is that there are not efficient data to prove the result and support the conclusion of this study, which made it less solid and significant.

Other comments:

1.     Please combine the images at the same position into one combined figure as Figure 1a, b,…

2.     Please carefully check your grammar and misspellings (e.g., line 124 NFC-tag ad Aerogel) before the next submission. 

3.     Please give the full name of the abbreviations at the first time they appear. (e.g., NFC) 

4.     Please add scale bar to each photo image.

5.     Some figures are unimportant and should be removed from the paper. Such as Figure 5, 11, 13

6.     The authors mentioned table 1 in line 241, but there is no table at all. 

7.     The figure caption of Figure 14 should be place at the bottom of the figure. 

Author Response

The authors acknowledge the useful observations and suggestions of the reviewer’s as concerns the manuscript entitled: 

Intelligent materials for labeling dentures in forensic dental identification

Corina Laura Åžtefănescu1,*, Lavinia Simona Neculai Cândea2, Marius Florentin Popa2, Agripina Zaharia1, Rodica Maria Murineanu1, Åžtefan Pricop2, Liliana Sachelarie3,*, Loredana Liliana Hurjui4, Vlad Danila4

According to the reviewer’s recommendations, all the suggestions were taken into account, as follows:

This paper aimed at providing a novel labeling method on dentures for individual identifications. This is an interesting and meaningful field which is not fully studied yet.   

However, the poor writing, paragraphing and figure layout made the manuscript hard to read. Major comments:

  1. I strongly recommend the authors reorganize the paragraphs and the figures, especially the method part. 

We reorganized the paragraphs and figures

  1. Another big flaw of this paper is that there are not efficient data to prove the result and support the conclusion of this study, which made it less solid and significant.

We tried to reproduce this method as better as possible and reorganized the material. Thank you!

Other comments:

  1. Please combine the images at the same position into one combined figure as Figure 1a, b,…

Done

  1. Please carefully check your grammar and misspellings (e.g., line 124 NFC-tag ad Aerogel) before the next submission. 

Done

  1. Please give the full name of the abbreviations at the first time they appear. (e.g., NFC) 

Done

  1. Please add scale bar to each photo image.

Done

  1. Some figures are unimportant and should be removed from the paper. Such as Figure 5, 11, 13

Done

  1. The authors mentioned table 1 in line 241, but there is no table at all. 

Done

  1. The figure caption of Figure 14 should be place at the bottom of the figure. 

Done

Thank you very much for your review,

Respectfully,

Prof.dr. Liliana Sachelarie

Reviewer 3 Report (New Reviewer)

Interesting work, but a few comments. All equipment and materials must be provided with the manufacturer and country of origin. Second, use impersonal forms.

Introduction

 Line 99

NFC-tags- If you use an abbreviation, state what it means the first time you use it

Line 124

We studied the properties- please use impersonal forms in the text -  In this work it was made, studied etc.

Aim of study should be placed in the introduction part.

In the introduction part, you could add some information about the acrylic itself and its wide use in prosthetics, e.g Raszewski Z, Nowakowska D, Wieckiewicz W, Nowakowska-Toporowska A. Release and Recharge of Fluoride Ions from Acrylic Resin Modified with Bioactive Glass. Polymers (Basel). 2021 Mar 27;13(7):1054. doi: 10.3390/polym13071054. PMID: 33801712; PMCID: PMC8037481

Materials and methods

Line 139

32 sample dentures measuring 2/4/0.2 cm

He would use the term rectangular samples, because the prosthesis has a definite shape, and only small slices of it were used here.

The second thing is the materials and their producers, heat curing acrylic resin (name, producer, country).

Line 142

Sinks, filament?-  this would require further clarification. . Why use hard plaster after the wax has been boiled? What is this filament?

Or to shorten it altogether and write that the samples were prepared in the traditional technique of hot acrylic polymerization and elimination of wax.

Line 150

patients allergic to the pink acrylate dye- red pigment,  In our experiments, we used- pleas used impersonal form

Aerogel in granular form- producer country

Line 163

We produced- impersonal form

Line 170

To make the mold we used flasks for investing the sample dentures- To make the mold denture flasks were used.

Line 171

we applied- impersonal form

line 173

cold water for 10 minutes for insulation-normally, separators are used here gypsum acrylic based on an aqueous solution of alginate or petroleum jelly, otherwise the two parts of the gypsum stick together

rinsing tank- producer country? Or it was boiled  simply.

 Isodent (Spofa Dental, Czech Republic)

Line 194

We proceeded to their rough processing with milling acrylic burs of different types and sizes, to remove the excess acrylic material and plaster. The smoothing and finishing was made with special rubbers-polypants of different grain widths- producers of polishing tools , country, impersonal form

Line 206

We included- impersonal form

Figures 7,8,9 captions under them should be smaller so that they do not merge into one whole

Don't use steps everywhere. According to me, as a practitioner, making dentures, we first use cutting tools, milling cutters, Fig. 7, powerful polishing silicone polisher, Fig. 9, and only at the  end, a felt wheel for polishing to a high gloss, Fig. 8.

Line 228

Thus, we obtained, we chose liquid materials

Line 240

One sample denture of each type was placed in each -don't use the same word twice in a sentence, English is a rich language. In each container, one tested sample was immersed.

According me Figures 10 - 123 are superfluous in this work.

 Samples were storage at room temperature during 4 years in closed container and in direct sun light exposition

Figure 14 and 15.

To prevent this type of error in the future, you should put acrylic samples in flasks and put PE foil between the two parts, top and bottom of the cuvette. Then put the whole thing in the press 10 minutes-15 minutes. Open the flask, remove the foil. Cut out a small amount of acrylic with a knife and place the microprocessor in the middle of pouch made in acrylic dough. Next, close the cuvette and press again. After 15 minutes, the acrylic will be hard enough that the microchip will not move. Next step polymerization.

The most interesting element is missing in the material and methods part. how will you use this insulating gel? Producing 5-layer samples. acrylic, gel, microchip, gel, acrylic. Because it will have the greatest practical meaning???

From line 289 these are the results. That's a new part of the article

I would also add a table to some of the results.

Table… The number of samples that could be read with the application, after 3 years stored in different environments

Sea water

Water

HCl acid

Acrylic/ microchip/ acrylic

4/4

4/4

2/4*

Acrylic/ aero gel

4/4

4/4

4/4

*Samples which are not readable because the microchip will move during the manufacture of the prosthesis

line 303 and 307. When giving reference, they should be given ascending, i.e. first 17-19 and then 20-21 and not 23-24.

 Line 318

Aerogel or Aerogel in combination with NFC-tag- Aerogel in combination with NFC-tag - The goal of your research was to determine if Aerogel is a good insulator for microchips, so stick with it.

Line 322

Our studies

Line 325

intelligent materials or "materials of the future" - this is a bit too intelligent material, a bit like from an advertising campaign. simply prostheses with a built-in information carrier

line 337

we noticed

line 338- I gave you the solution

line 345

Last but not least,- common unprofessional language

Line 348

by us for-

conclusion

  could be given in the form:

·         the “sandwich” technique can improve   the labeling materials in the dentures, and their limitation to the previous methods related to discomfort and lack of esthetics from the patient`s.

·         The effectiveness of labeling with NFC-tags alone or with NFC-tags embedded in Aerogel. The information was preserved both during the polymerization process as well as for a long time afterwards, despite exposure to environments that usually damage or destroy other types of labeling

The last paragraph is not a conclusion. The positive results motivate us to expand our experiments in the sense of using…..

good luck with further research

Author Response

The authors acknowledge the useful observations and suggestions of the reviewer’s as concerns the manuscript entitled: 

Intelligent materials for labeling dentures in forensic dental identification

Corina Laura Åžtefănescu1,*, Lavinia Simona Neculai Cândea2, Marius Florentin Popa2, Agripina Zaharia1, Rodica Maria Murineanu1, Åžtefan Pricop2, Liliana Sachelarie3,*, Loredana Liliana Hurjui4, Vlad Danila4

According to the reviewer’s recommendations, all the suggestions were taken into account, as follows:

Interesting work, but a few comments. All equipment and materials must be provided with the manufacturer and country of origin. Second, use impersonal forms.

DONE

Introduction

 Line 99

NFC-tags- If you use an abbreviation, state what it means the first time you use it

Done

Line 124

We studied the properties- please use impersonal forms in the text -  In this work it was made, studied etc.

Done

Aim of study should be placed in the introduction part.

Done

In the introduction part, you could add some information about the acrylic itself and its wide use in prosthetics, e.g. doi: 10.3390/polym13071054. PMID: 33801712; PMCID: PMC8037481

Done

Materials and methods

Line 139

32 sample dentures measuring 2/4/0.2 cm

He would use the term rectangular samples, because the prosthesis has a definite shape, and only small slices of it were used here.The second thing is the materials and their producers, heat curing acrylic resin (name, producer, country).

Done

Line 142

Sinks, filament?-  this would require further clarification. . Why use hard plaster after the wax has been boiled? What is this filament?

Or to shorten it altogether and write that the samples were prepared in the traditional technique of hot acrylic polymerization and elimination of wax.

Done

Line 150

patients allergic to the pink acrylate dye- red pigment,  In our experiments, we used- pleas used impersonal form

Aerogel in granular form- producer country

Done

Line 163

We produced- impersonal form

Done

Line 170

To make the mold we used flasks for investing the sample dentures- To make the mold denture flasks were used.

Done

Line 171

we applied- impersonal form

Done

line 173

cold water for 10 minutes for insulation-normally, separators are used here gypsum acrylic based on an aqueous solution of alginate or petroleum jelly, otherwise the two parts of the gypsum stick together

rinsing tank- producer country? Or it was boiled  simply.

 Isodent (Spofa Dental, Czech Republic)

MODIFIED

Line 194 - Done

We proceeded to their rough processing with milling acrylic burs of different types and sizes, to remove the excess acrylic material and plaster. The smoothing and finishing was made with special rubbers-polypants of different grain widths- producers of polishing tools , country, impersonal form

Line 206- Done

We included- impersonal form

Figures 7,8,9 captions under them should be smaller so that they do not merge into one whole

Don't use steps everywhere. According to me, as a practitioner, making dentures, we first use cutting tools, milling cutters, Fig. 7, powerful polishing silicone polisher, Fig. 9, and only at the  end, a felt wheel for polishing to a high gloss, Fig. 8.

Line 228- Done

Thus, we obtained, we chose liquid materials

Line 240 - Done

One sample denture of each type was placed in each -don't use the same word twice in a sentence, English is a rich language. In each container, one tested sample was immersed.

According me Figures 10 - 123 are superfluous in this work.

 Samples were storage at room temperature during 4 years in closed container and in direct sun light exposition

Figure 14 and 15.

To prevent this type of error in the future, you should put acrylic samples in flasks and put PE foil between the two parts, top and bottom of the cuvette. Then put the whole thing in the press 10 minutes-15 minutes. Open the flask, remove the foil. Cut out a small amount of acrylic with a knife and place the microprocessor in the middle of pouch made in acrylic dough. Next, close the cuvette and press again. After 15 minutes, the acrylic will be hard enough that the microchip will not move. Next step polymerization.

The most interesting element is missing in the material and methods part. how will you use this insulating gel? Producing 5-layer samples. acrylic, gel, microchip, gel, acrylic. Because it will have the greatest practical meaning???

Thank you for the advice and method. For our next research, we will make it like you suggested.

We detailed the sandwich technique after your evaluation.

From line 289 these are the results. That's a new part of the article

I would also add a table to some of the results.

Table… The number of samples that could be read with the application, after 3 years stored in different environments

Sea water

Water

HCl acid

Acrylic/ microchip/ acrylic

4/4

4/4

2/4*

Acrylic/ aero gel

4/4

4/4

4/4

Table 2. Number of samples read (with the NFC reader application) after 3 months storage in different environments

Sample type

Liquid media

Sea  water

Fresh water

Alcohol (40%)

HCl (0.2%)

Acrylic layer / NFC-tag / acrylic layer / PINK

4/4

4/4

4/4

2*/4

Acrylic layer / NFC-tag /acrylic layer / TRANSPARENT

4/4

4/4

4/4

4/4

Acrylic layer / Aerogel / NFC-tag / Aerogel / acrylic layer / PINK

4/4

4/4

4/4

2*/4

Acrylic layer / Aerogel / NFC-tag / Aerogel / acrylic layer / TRANSPARENT

4/4

4/4

4/4

4/4

Note: * Samples are not readable due to an error in the technological stage of manufacturing

*Samples which are not readable because the microchip will move during the manufacture of the prosthesis

line 303 and 307. When giving reference, they should be given ascending, i.e. first 17-19 and then 20-21 and not 23-24.

Done

 Line 318- Done

Aerogel or Aerogel in combination with NFC-tag- Aerogel in combination with NFC-tag - The goal of your research was to determine if Aerogel is a good insulator for microchips, so stick with it.

Line 322 - Done

Our studies

Line 325- Done

intelligent materials or "materials of the future" - this is a bit too intelligent material, a bit like from an advertising campaign. simply prostheses with a built-in information carrier

line 337 - Done

we noticed

line 338- I gave you the solution – Thank you!

line 345 - Done

Last but not least,- common unprofessional language

Line 348 - Done

by us for-

conclusion

  could be given in the form:

  • the “sandwich” technique can improve   the labeling materials in the dentures, and their limitation to the previous methods related to discomfort and lack of esthetics from the patient`s.
  • The effectiveness of labeling with NFC-tags alone or with NFC-tags embedded in Aerogel. The information was preserved both during the polymerization process as well as for a long time afterwards, despite exposure to environments that usually damage or destroy other types of labelin

The last paragraph is not a conclusion. The positive results motivate us to expand our experiments in the sense of using…..

Done

good luck with further research

Thank you very much for your review,

Respectfully,

Prof.dr. Liliana Sachelarie

Reviewer 4 Report (Previous Reviewer 3)

It was evaluated the article titled “Intelligent materials for labeling dentures in forensic dental identification”.

The goal of this article was “to study the properties of intelligent materials (NFC-tags ad Aerogel) and tested them for their ability to preserve certain identification data over time unaltered, taking advantage of the Aerogel insulation ability”.

The topic is interesting, and the content interesting. Therefore, some concerns were raised.

ABSTRACT

Methods and results were poorly presented. Please, improve these parts.

INTRO: I considered a little long. Shorten it.

lines 44-52; 54-56; 60-64, 111-116 : include a refs.

M&M: this section is complete but the language used is not scientific. Please, rewrite it with all content presented with more scientific language.

- subtopic 2.1 can be moved to the end of the intro.

- line 139: “a number of 32 sample dentures”. why this number?

- was there any statistical analysis?

CONCLUSION

“They provide additional reliability in odontological identification compared to other usual forensic techniques.” How can the authors conclude it?

Author Response

The authors acknowledge the useful observations and suggestions of the reviewer’s as concerns the manuscript entitled: 

Intelligent materials for labeling dentures in forensic dental identification

Corina Laura Åžtefănescu1,*, Lavinia Simona Neculai Cândea2, Marius Florentin Popa2, Agripina Zaharia1, Rodica Maria Murineanu1, Åžtefan Pricop2, Liliana Sachelarie3,*, Loredana Liliana Hurjui4, Vlad Danila4

According to the reviewer’s recommendations, all the suggestions were taken into account, as follows:

It was evaluated the article titled “Intelligent materials for labeling dentures in forensic dental identification”.

The goal of this article was “to study the properties of intelligent materials (NFC-tags ad Aerogel) and tested them for their ability to preserve certain identification data over time unaltered, taking advantage of the Aerogel insulation ability”.

The topic is interesting, and the content interesting. Therefore, some concerns were raised.

ABSTRACT- Done

Methods and results were poorly presented. Please, improve these parts.

 Done

INTRO: I considered a little long. Shorten it.

lines 44-52; 54-56; 60-64, 111-116 : include a refs.

 Done

M&M: this section is complete but the language used is not scientific. Please, rewrite it with all content presented with more scientific language.

- subtopic 2.1 can be moved to the end of the intro.

- line 139: “a number of 32 sample dentures”. why this number?

This research is a pilot study. The 32 sample dentures were chosen according to the materials used and the 4 liquid environments in which each of them were immersed, respectively:8 control sample dentures (4 pink acrylate and 4 transparent acrylate)8 sample dentures with NFC-tag alone (4 pink acrylate and 4 transparent acrylate)8 sample dentures with Aerogel alone (4 pink acrylate and 4 transparent acrylate)8 sample dentures with NFC-tag +Aerogel (4 pink acrylate and 4 transparent acrylate)

- was there any statistical analysis?

Statistical analysis requires a large number of samples and variables. In this case it could not be done.

CONCLUSION

“They provide additional reliability in odontological identification compared to other usual forensic techniques.” How can the authors conclude it?

All your observations can be found in the new version. After the evaluation of all reviewers, the article was completely redone.

According to the reviewer’s recommendations, all the suggestions were taken into account, as follows:

Thank you very much for your review,

Respectfully,

Prof.dr. Liliana Sachelarie

Round 2

Reviewer 2 Report (New Reviewer)

This version has been improved a lot in readability. Though it still has flaws in the small sample number and not enough data presenting, but this paper gives a new view of labeling dentures and worth publishing. 

Author Response

The authors acknowledge the useful observations and suggestions of the reviewer’s as concerns the manuscript entitled

Intelligent materials for labeling dentures in forensic dental identification- a pilot study

Corina Laura Åžtefănescu1,*, Lavinia Simona Neculai Cândea2, Marius Florentin Popa2, Agripina Zaharia1, Rodica Maria Murineanu1, Åžtefan Pricop2, Liliana Sachelarie3,*, Loredana Liliana Hurjui4, Vlad Danila4

According to the reviewer’s recommendations, all the suggestions were taken into account, as follows: 

Thank you very, very much for your support and encouragement for future research!!!!

Respectfully,

Prof.dr. Liliana Sachelarie

Reviewer 4 Report (Previous Reviewer 3)

It was evaluated the article titled “Intelligent materials for labeling dentures in forensic dental identification”.

The goal of this article was “to study the properties of intelligent materials (NFC-tags ad Aerogel) and tested them for their ability to preserve certain identification data over time unaltered, taking advantage of the Aerogel insulation ability”.

The authors said in the response that it is a pilot study. Include this info in the title.

ABSTRACT: improved but the conclusion is bigger than the results.

Or were results not important?

INTRO: I requested to “I considered a little long. Shorten it.”

the authors said “done”. When I read, the intro was bigger.

M&M:

my question: “was there any statistical analysis?”

authors’ response: “Statistical analysis requires a large number of samples and variables. In this case it could not be done.”

This response is completely incorrect. What is the aim to do a pilot study then?

Author Response

The authors acknowledge the useful observations and suggestions of the reviewers as concerns the manuscript entitled

Intelligent materials for labeling dentures in forensic dental identification- a pilot study

Corina Laura Åžtefănescu1,*, Lavinia Simona Neculai Cândea2, Marius Florentin Popa2, Agripina Zaharia1, Rodica Maria Murineanu1, Åžtefan Pricop2, Liliana Sachelarie3,*, Loredana Liliana Hurjui4, Vlad Danila4

According to the reviewer’s recommendations, all the suggestions were taken into account, as follows: 

  1. The authors said in the response that it is a pilot study. Include this info in the title.

Done

Intelligent materials for labeling dentures in forensic dental identification- a pilot study

  1. ABSTRACT: improved but the conclusion is bigger than the results. Or were results not important?

Thank you very much!

(3) Results: The information stored on NFC tags could be retrieved unaltered at the end of the monitoring period, regardless of the damage caused by the liquid media to the sample dentures material, and provide an innovative solution as compared to other labeling methods, from a forensic, social, and legal point of view. (4) Conclusions: The use of intelligent materials for labeling acrylic dentures provides additional reliability by preserving the identification information in time. 

INTRO: I requested to “I considered a little long. Shorten it.”

the authors said, “done”. When I read, the intro was bigger. 

It's our fault. We apologize the introduction is redone according to the requirements. You can see in the paper.

M&M:

my question: “Was there any statistical analysis?”

authors’ response: “Statistical analysis requires a large number of samples and variables. In this case, it could not be done.”

This response is completely incorrect. What is the aim to do a pilot study then?

Honestly, we didn't think about doing a statistical analysis. Thank you for the suggestion and we will use the statistical analysis in the following research.

Thank you very much for your review,

Respectfully,

Prof. dr. Liliana Sachelarie

This manuscript is a resubmission of an earlier submission. The following is a list of the peer review reports and author responses from that submission.

Round 1

Reviewer 1 Report

Interesting text, touching on a new field of science. However, it must be written integrally and independently. Many parts of the text are copied word for word from sources available on the Internet. In this form, the publication may not be approved for use.

Reviewer 2 Report

Accept in present form.

Reviewer 3 Report

I evaluated the article “Intelligent materials for labeling dentures in forensic dental identification", with the goal: ???

ABSTRACT

- describe better the: GOAL, METHODS, RESULTS

INTRO - I considered this section long. It could be reduced with a better conection between paragraphs. Review the text.

- Lines 53 and 57 started with “Forensic odontology”. Change it

- The aim of the study was not inserted in the text.

METHODS

- was there any statistical analysis?

- How the authors compared the samples?

RESULTS: this section was poorly described. Improve it!

DISCUSSION: I suggest to improve this part.

CONCLUSION: there is questions about the conclusion due to the M&M used.

Reviewer 4 Report

The manuscript presents marking techniques by using smart materials (NFC tags, Aerogel). The test is simple and the manuscript lacks deep analysis and discussion. More characteristics should be carried out and analysis and discussion should be performed in detail. The manuscript does not include the important findings and scientific inclusion. The English is poor and SHOULD be improved substantially. Accordingly, it is not like an academic paper and not suitable to be published in the journal of Applied Sciences.